# Oropouche infection in Peruvian patients: A systematic review and meta-analysis

**Darwin A. León-Figueroa[1,2], Edwin Aguirre-Milachay[3], Milagros Diaz-Torres[1], Jean Pierre Villanueva-De La Cruz[4], Edwin A. Garcia-Vasquez[1], Virgilio E. Failoc-Rojas[5], Mario J. Valladares-Garrido[6]\***

1 Facultad de Medicina Humana, Universidad de San Martín de Porres, Chiclayo, Peru, 2 EpiHealth Research Center for Epidemiology and Public Health, Lima, Peru, 3 Hospital Nacional Almanzor Aguinaga Asenjo, Chiclayo, Peru, 4 Facultad de Medicina de la Universidad Nacional de Trujillo, La Libertad, Perú, 5 Universidad San Ignacio de Loyola, Lima, Peru, 6 Escuela de Medicina Humana, Universidad Señor de Sipán, Chiclayo, Peru

\* vgarrido@uss.edu.pe

## Abstract

### Background

The Oropouche virus (OROV), discovered in 1955, has evolved from being a pathogen limited to the Amazon basin to becoming a growing threat to public health in Latin America. Because its symptoms are similar to those of dengue and zika, diagnosis is complicated. In this context, the objective of this study is to determine the prevalence of epidemiological and clinical characteristics in Peruvian patients diagnosed with Oropouche.

### Methods

A systematic review and meta-analysis were performed in accordance with PRISMA guidelines. An exhaustive literature search was conducted up to April 10, 2025, across ten databases using MeSH terms like "Oropouche" and "Peru," combined with Boolean operators. Only observational studies conducted in Peru that reported confirmed OROV infections through reverse transcription polymerase chain reaction (RT-PCR) or enzyme-linked immunosorbent assay (ELISA), and that described clinical or epidemiological characteristics, were included. The methodological quality of these studies was evaluated using the JBI-MAStARI tool. To estimate the pooled prevalence and 95% confidence intervals, random-effects models were applied in R (version 4.2.3). Heterogeneity was assessed using the I² statistic, and publication bias was evaluated through funnel plots and Egger's test, when applicable.

### Results

Six observational studies published between 2010 and 2020 were included, involving 396 Peruvian patients diagnosed with OROV by RT-PCR or ELISA. The studies were

**Data availability statement:** All relevant data are within the paper and its Supporting information files.

**Funding:** The author(s) received no specific funding for this work.

**Competing interests:** The authors have declared that no competing interests exist.

conducted in Piura, Loreto, Huánuco, Madre de Dios, and San Martín. Most patients were between 20 and 30 years old; 44.9% were male. All studies were of moderate quality. Due to the limited number of studies, publication bias was not assessed. The most common symptoms were fever, headache, myalgia, arthralgia, and retro-ocular pain.

## Conclusion

The findings of this study reveal a significant occurrence of diverse symptoms in Peruvian patients infected with OROV. Due to the clinical resemblance to other arbo-viruses, it is essential to establish more precise diagnostic methods to prevent misdi-agnosis and underreporting. The existing evidence remains limited, highlighting the importance of enhancing epidemiological monitoring, improving diagnostic tools, and creating public health strategies specifically targeted at endemic regions to reduce the effects of this emerging infection.

## 1. Introduction

The Oropouche virus (OROV), belonging to the *Peribunyaviridae* family and the *Orthobunyavirus* genus, has emerged as an increasing public health challenge in Latin America and the Caribbean [1]. Its discovery dates back to 1955 in Trinidad, when it was first identified from a blood sample of a forestry worker who had a fever [2]. Since then, multiple epidemic outbreaks have been reported in several South American countries, including Brazil, Peru, Bolivia, Colombia, Ecuador, Panama, and more recently, Cuba, where its presence was confirmed for the first time in 2024 [3].

The OROV causes an acute febrile illness known as Oropouche fever, character-ized by high fever, intense headache, myalgia, arthralgia, photophobia, and general malaise [4]. In recent years, severe cases with neurological manifestations, including meningitis, encephalitis, Guillain-Barré syndrome, and even microcephaly associ-ated with vertical transmission, have been documented, expanding the understand-ing of the clinical spectrum of this disease [5]. Moreover, the increasing number of severe neurological cases highlights the need for enhanced clinical surveillance and specialized care in affected regions. Furthermore, fatal cases of OROV have been documented in regions outside its usual endemic areas, particularly in Brazil. This represents the first recorded instances of such deaths, signaling a notable increase in the severity of the virus's impact. This situation has sparked concerns regarding the potential emergence of more virulent strains, underscoring the urgent need for enhanced monitoring and immediate public health interventions to prevent additional fatalities [6].

Transmission of the virus primarily occurs through the bite of female Culicoides mosquitoes (particularly *Culicoides paraensis*), although other vectors, such as *Aedes aegypti, Aedes albopictus,* and *Culex quinquefasciatus*, have also been shown to transmit the virus [7,8]. This diversity of vectors increases the risk of geographical expansion, particularly in urban areas with high population density [9]. Importantly,

there is growing evidence of an overlap between vectors of OROV and those of other mosquito-borne diseases like dengue, which can complicate diagnosis and influence symptom presentation [4,10].

In Peru, various arboviral diseases, such as Oropouche, dengue, Zika, chikungunya, and yellow fever, circulate, sharing similar clinical manifestations that can complicate diagnosis. The most prevalent disease is dengue, which occurs in all 24 regions of the country (Fig 1) [11]. All four serotypes of the dengue virus (DENV-1, DENV-2, DENV-3, and DENV-4) have circulated in Peru, with DENV-1 and DENV-2 being the most common [12,13].

However, in secondary infections, the DENV-2 serotype has been associated with more severe clinical outcomes and higher mortality [14–16]. On the other hand, primary infections with DENV-3 have shown a higher proportion of symptoms and complications, often presenting a potentially more aggressive clinical profile [17,18]. Recent studies indicate that

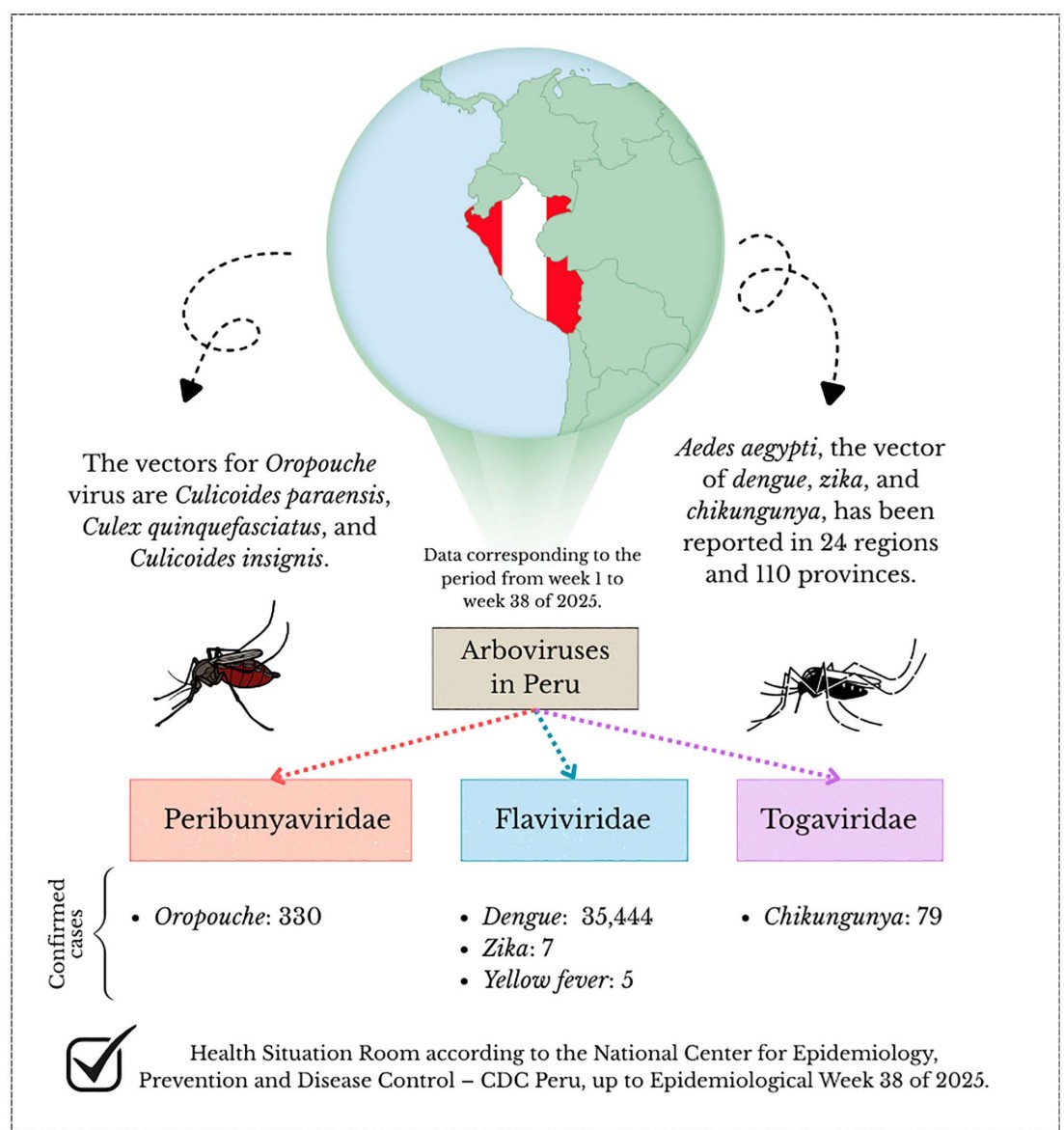

**Fig 1. Scope of arbovirus (dengue, zika, chikungunya, and Oropouche) in Peruvian patients according to the CDC Peru.**

co-circulation of OROV with other arboviruses, especially dengue, can lead to an increase in misdiagnoses, particularly as the clinical manifestations of these diseases overlap, such as fever, headache, and myalgia [19,20].

According to the Pan American Health Organization (PAHO), as of epidemiological week 38 of 2025, Peru has reported 330 confirmed cases of OROV, ranking as the third country with the most cases in the Americas, after Brazil and Panama [21]. In recent years, a significant resurgence of the virus has been observed, with a record number of confirmed cases in the region during the 2023–2025 period, primarily in Brazil, Peru, Bolivia, Cuba, and Panama [22]. This resurgence is partially attributed to the increased vector population, enhanced by climate factors such as rising temperatures and changing precipitation patterns. Moreover, human behavior and migration also play critical roles in the spread of the disease [23]. This increase is likely compounded by the enhanced vector competence of *Culicoides* species, with studies showing that changes in environmental conditions, such as increased rainfall and temperature, have created more favorable conditions for the vectors' breeding and survival [22,24]. This highlights the intricate interplay between climatic factors, vector ecology, and public health responses in the region [25,26].

Given the rapid increase in case incidence, geographic expansion, and the severity of certain clinical forms, the OROV constitutes an emerging disease that requires priority attention in public health [27]. The PAHO and the European Centre for Disease Prevention and Control (ECDC) have issued epidemiological alerts and provisional guidelines to strengthen entomological surveillance, improve early diagnosis, and promote integrated prevention strategies [28,29]. Nevertheless, gaps in healthcare access and public awareness continue to hinder effective control, and more targeted interventions are necessary [27,29].

In this context, the development of the present study, which comprehensively addresses the epidemiological and clinical aspects of OROV in Peru, is crucial for improving timely diagnosis, optimizing clinical management, and guiding the design of surveillance and control strategies tailored to the national context. Furthermore, this initiative would help update and consolidate existing knowledge, close gaps in the scientific literature, strengthen the public health response, and guide future research in the region [30].

## 2. Materials and methods

### 2.1. Protocol and registration

The study was conducted in accordance with the Preferred Reporting Items for Systematic Reviews and Meta-Analyses (PRISMA) guidelines (S1 Table) [31]. Additionally, the research protocol was registered with the Prospective International Registry of Systematic Reviews (PROSPERO) under identification number CRD42024594712. The initially registered protocol underwent moderate modifications, as the researchers decided to focus on the Peruvian population. Consequently, adjustments were made to several sections, including the research question, literature search strategy, and eligibility criteria. Finally, meta-analysis studies focusing on disease characteristics were used as references to enrich our methodology and results approach [10,32,33]. This approach ensures that the systematic review is more accurately tailored to the target population while maintaining transparency and rigor in the research process.

### 2.2. Eligibility criteria

Eligible studies for this research had to meet the following criteria [10,32,33]: (1) confirmation of OROV infection through polymerase chain reaction (RT-PCR) or enzyme-linked immunosorbent assay (ELISA), (2) detailed descriptions of clinical manifestations and epidemiological characteristics of patients, (3) inclusion of observational studies such as cohort, case-control, cross-sectional studies, and case series, both prospective and retrospective, and (4) studies conducted on Peruvian patients. Excluded studies included: (1) those with a sample size smaller than five, (2) those involving non-human hosts, and (3) non-original research such as narrative and systematic reviews, meta-analyses, editorials, letters to the editor, conference abstracts, and randomized clinical trials.

## 2.3. Information sources and search strategy

A comprehensive search was conducted across ten electronic databases or search tools, both regional and international, including PubMed, Scopus, Web of Science, Embase, ScienceDirect, Google Scholar, Virtual Health Library, Scielo, Dimensions, and Epistemonikos, up to April 10, 2025, with no restrictions on language or time. The search strategy, which employed MeSH terms such as "Oropouche" and "Peru" combined with the logical operators AND and OR, was independently validated by two authors and is detailed in S2 Table. In addition to the main search, complementary methods were employed, such as manual searches in national journals and reviewing the reference lists of selected studies. Only peer-reviewed articles were considered to ensure the quality of the included studies. These articles were selected using controlled vocabulary and keywords related to the OROV. Potential studies were reviewed to verify their alignment with the criteria set out in the main search strategy [32,33].

## 2.4. Study selection

The search results were stored using EndNote software, version X9 (Thomas Reuters, New York, NY, USA). Duplicated articles were then removed. The remaining titles and abstracts were reviewed independently. The full articles were subsequently thoroughly examined to verify compliance with the inclusion criteria. Any discrepancies arising during this process were resolved through consensus in a meeting [32,33].

## 2.5. Quality assessment

The quality and possible biases of the studies included in the meta-analysis were assessed using the Joanna Briggs Institute Meta-Analysis of Statistics Assessment and Review Instrument (JBI-MAStARI). According to their scores, the studies were classified into three quality tiers: high quality (≥ 7 points), moderate quality (4–6 points), and low quality (< 4 points) (S3 Table) [32,33].

## 2.6. Data collection process and data items

The data from the articles were systematically organized into an Excel spreadsheet. Two authors independently and manually extracted a comprehensive dataset, which included the following variables: authors, year of publication, study design, geographical region, sample characteristics, gender (male/female), age (in years), Oropouche diagnostic method, study site, data collection methods, disease progression, and clinical features (fever, headache, myalgia, arthralgia, anorexia/hyporexia, retroocular pain, abdominal pain, nausea/vomiting, diarrhea, chills, lumbar pain, odynophagia, cutaneous rash, conjunctival injection, petechiae, and cough).

In a subsequent meeting, the two authors compared their data extractions, and any discrepancies were resolved through consensus. A third independent investigator then conducted a thorough review and verification process to ensure the accuracy and integrity of the extracted data.

## 2.7. Data analysis

A meta-analysis of prevalence (proportions) was performed using R software version 4.2.3 (https://www.r-project.org/) (S5 and S6 Tables). To estimate the combined prevalence of epidemiological and clinical characteristics in Peruvian patients with Oropouche, a variance-weighted inverse random effects model was applied. The variability between studies was evaluated using the Cochrane Q statistic, while heterogeneity was quantified using the Inconsistency Index ($I^2$). Heterogeneity levels were categorized as low (<25%), moderate (25%–50%), and high (>75%) [10,32,33].

To assess potential publication bias, two methods were employed: visual inspection of the funnel plot and Egger's test. These tests were applied only when the meta-analysis included at least 10 studies, as fewer studies reduce the ability to detect true asymmetry. Publication bias was considered significant if the p-value was below 0.05 [32,33].

The study's results were presented in tables and descriptive graphs. A forest plot was used to visually display the combined prevalence of epidemiological and clinical characteristics among Peruvian patients with Oropouche, including 95% confidence intervals to provide a more accurate representation of the data [32,33].

## 3. Results

### 3.1. Study selection

The search strategy across ten electronic databases and search tools yielded a total of 152 articles. After removing duplicates (n = 105), 47 articles were selected by comparing their titles and abstracts against the inclusion criteria. A thorough evaluation of 14 full-text articles was then conducted, leading to the final inclusion of 6 studies in the systematic review and meta-analysis (S5 Table) [34–39]. The selection process is visually summarized in Fig 2 using a PRISMA flow diagram.

### 3.2. Characteristics of the included studie

Six observational studies published between 2010 and 2020 were included, involving a total of 396 patients diagnosed with Oropouche through RT-PCR, ELISA IgM, and/or IgG tests. These studies were conducted in various regions of Peru, including Piura, Loreto, Huánuco, Madre de Dios, and San Martín. The sample sizes varied, ranging from 12 to 131 patients, all with favorable outcomes (Table 1) [34–39].

Regarding the gender distribution, 44.9% of the patients (178/308) were male, and 32.8% (130/308) were female. The most frequent age range of the diagnosed patients was between 20 and 30 years old (Table 1) [34–39].

### 3.3. Quality of the included studies and publication bias

The studies included in the analysis were deemed of moderate quality, as outlined in S3 Table. Due to the inclusion of fewer than 10 studies in the meta-analysis, assessing publication bias via funnel plot visualization or the Egger test was not feasible [34–39].

### 3.4. Prevalence of clinical characteristics of Peruvian patients with Oropouche

The distribution of the prevalence of clinical manifestations in Peruvian patients with Oropouche (Table 2) [34–39], organized from highest to lowest prevalence, included fever (99%, 95% CI: 97–100%) [34–39], headache (86%, 95% CI: 80–91%) [34–39], myalgia (77%, 95% CI: 71–83%) [34,36–39], arthralgia (70%, 95% CI: 63–76%) [34–39], anorexia/ hyporexia (56%, 95% CI: 45–67%) [34,36–39], retroocular pain (51%, 95% CI: 40–62%) [34–39], nausea/ vomiting (45%, 95% CI: 39–52%) [35–39], lumbar pain (47%, 95% CI: 37–58%) [36,37,39], chills (40%, 95% CI: 0–96%) [35,37–39], abdominal pain (8%, 95% CI: 0–24%) [36–39], odynophagia (21%, 95% CI: 7–40%) [34,36–39], diarrhea (15%, 95% CI: 5–30%) [35,38,39], cutaneous rash (14%, 95% CI: 4–26%) [34,35,37–39], cough (10%, 95% CI: 0–38%) [35,36,38,39], conjunctival injection (9%, 95% CI: 0–50%) [36,38,39], and petechiae (1%, 95% CI: 0–2%) [34,36,37,39] (Table 3 and Fig 3).

## 4. Discussion

This work compiles information from six observational studies conducted in various regions of the country over a span of more than ten years. The findings provide a solid foundation for understanding the clinical and epidemiological characteristics of this arboviral disease, whose presence has gained increasing significance in the South American region due to its expanding geographical spread.

The evidence suggests a high prevalence of characteristic clinical symptoms, such as fever, headache, myalgia, arthralgia, and retroocular pain, in Peruvian patients with OROV, which align with the clinical patterns identified in Oropouche outbreaks in the Amazonian regions of countries such as Brazil, Venezuela, and Peru [36,40,41]. These figures

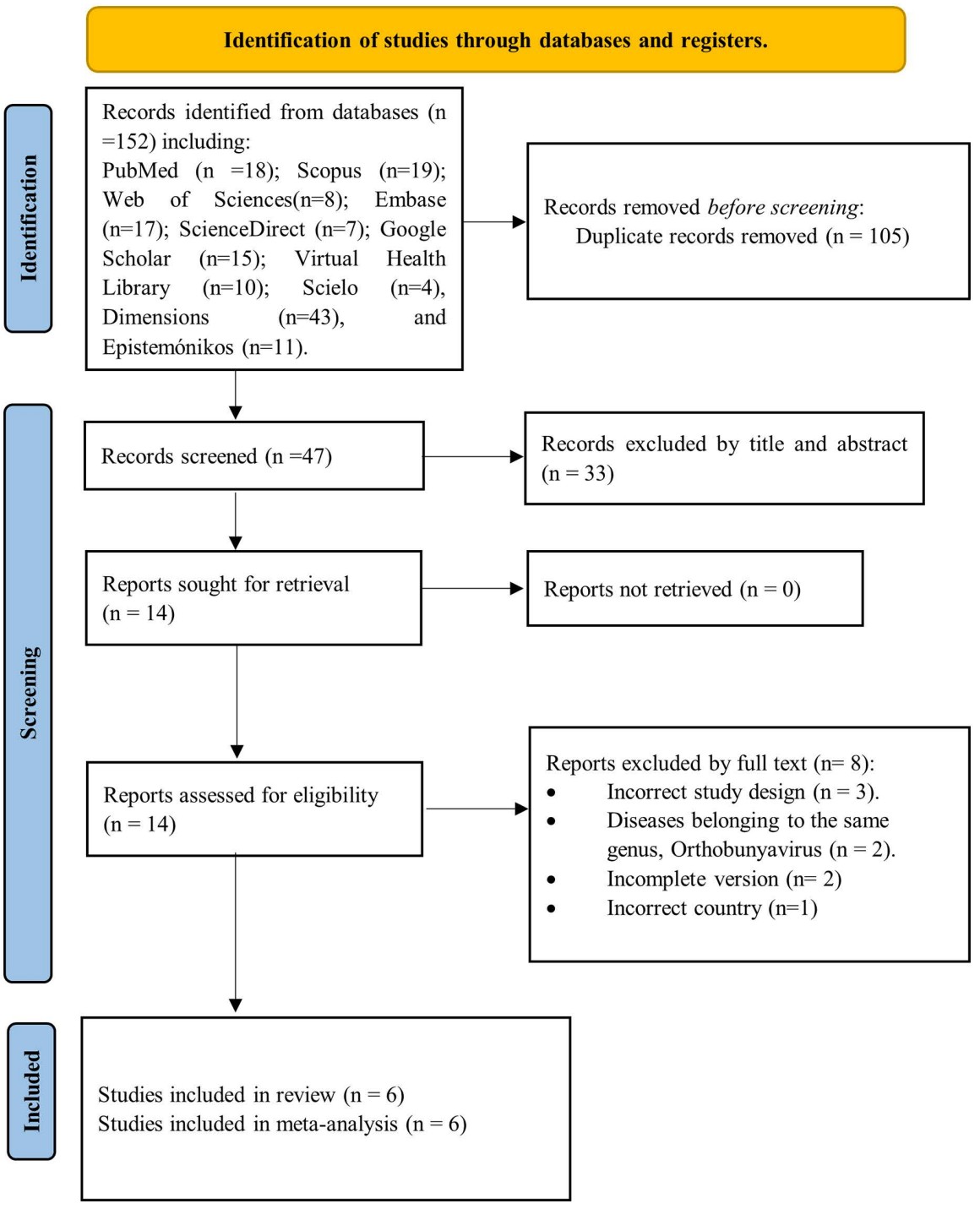

**Fig 2. Study selection process based on the PRISMA flowchart.**

Table 1. Summary of the epidemiological characteristics of the studies included in the meta-analysis.

| Authors | Year | Studio Type | Geographical region | Sample | Gender Male | Gender Female | Age (years) | Oropouche diagnostic method | Center where the study was conducted | Data collection methods | Evolution |
|---|---|---|---|---|---|---|---|---|---|---|---|
| Durango-Chavez HV, et al. [34] | 2022 | Retrospective cohort | Peru | 97 | 54 (55.7%) | 43 (44.3%) | <15: 39 15–34: 25 35–54: 17 ≥55: 16 | RT-PCR | The National Center of Epidemiology, Prevention of Disease Control of the Ministry of Health of Peru. | January 2015 to December 2016 | Recovered |
| Watts DM, et al. [35] | 2022 | Retrospective cohort | Loreto | 68 | 44 (64.7%) | 24 (35.3%) | 1–14: 8 15–29: 27 30–44: 14 >44: 18 | ELISA IgM and/or IgG | The Peruvian Ministries of Health (MOH) and Defense, Iquitos, Peru; and the University of Texas Medical Branch (UTMB), Galveston, Texas. | October 1, 1993 through September 30, 1999 | Recovered |
| Martins-Luna J, et al. [36] | 2020 | Retrospective cohort | Piura | 131 | 72 (55.0%) | 59 (45.0%) | <5: 8 (6.1%) 5–11: 26 (19.8%) 12–17: 13 (9.9%) 18–39: 33 (25.2%) 40–59: 22 (16.8%) ≥60: 29 (22.1%) | RT-PCR | National Center for Epidemiology, Disease Prevention, and Control of Peru. | February to September 2016 | Recovered |
| Silva-Caso W, et al. [37] | 2019 | Cross-sectional | Huanuco | 46 | NS | NS | Mean: 22.3 ± 15.6 | RT-PCR | The Leoncio Prado Health Network of the Ministry of Health of Peru and in the "Tingo María Contingency Hospital" | January and July 2016 | Recovered |
| Alva-Urcia C, et al. [38] | 2017 | Cross-sectional | Madre de Dios | 12 | 8 (66.7%) | 4 (33.3%) | Range: 5 - ≥45 | RT-PCR | Centro de Salud Nuevo Milenio, Centro de Salud Santa Rosa, Centro de Salud Laberinto, Centro de Salud La Joya, Centro de Salud Bélgica, Centro de Salud Iñapari, Centro de Salud Jorge Chávez, Centro de Salud El Triunfo and Centro de Salud Tres Islas. | January to March 2016 | Recovered |
| Alvarez-Falconi P, et al. [39] | 2010 | Retrospective cohort | San Martin | 42 | NS | NS | NS | ELISA IgM | Bagazan Villagers | April to May 2010 | Recovered |

NS: Not specified.

Table 2. Summary of the clinical characteristics of the studies included in the meta-analysis.

| Authors | Year | Fever | Headache | Myalgia | Arthralgia | Anorexia/Hyporexia | Retroocular pain | Abdominal pain | Nausea/Vomiting | Diarrhea | Chills | Lumbar pain | Odynophagia | Cutaneous rash | Conjunctival injection | Petechiae | Cough |
|---|---|---|---|---|---|---|---|---|---|---|---|---|---|---|---|---|---|
| Durango-Chavez HV, et al. [34] | 2022 | 97 (100%) | 82 (84.5%) | 75 (77.3%) | 63 (64.9%) | 62 (63.9%) | 52 (53.6%) | NR | NR | NR | NR | NR | 43 (44.3%) | 24 (24.7%) | NR | 1 (1%) | NR |
| Watts DM, et al. [35] | 2022 | 66 (98.5%) | 63 (94.0%) | NR | 56 (83.6%) | NR | 46 (68.7%) | NR | 35 (52.2%) | 18 (26.9%) | 61 (91.0%) | NR | NR | 10 (14.9%) | NR | NR | 16 (23.9%) |
| Martins-Luna J, et al. [36] | 2020 | 131 (100%) | 112 (85.5%) | 106 (80.9%) | 95 (72.5%) | 89 (67.9%) | 70 (53.4%) | 4 (3.1%) | 62 (47.3%) | NR | 0 (0%) | 66 (50.4%) | 48 (36.6%) | NR | 1 (0.8%) | 1 (0.8%) | 0 (0%) |
| Silva-Caso W, et al. [37] | 2019 | 46 (100%) | 35 (76.1%) | 35 (76.1%) | 30 (65.2%) | 23 (50%) | 28 (60.9%) | 4 (8.7%) | 22 (47.8%) | NR | NR | 16 (34.8%) | 16 (34.8%) | 15 (32.6%) | NR | 0 (0%) | NR |
| Alva-Urcia C, et al. [38] | 2017 | 12 (100%) | 8 (66.7%) | 6 (50%) | 7 (58.3%) | 4 (33.3%) | 4 (33.3%) | 0 (0%) | 3 (25%) | 1 (8.3%) | 3 (25%) | NR | 0 (0%) | 0 (0%) | 0 (0%) | NR | 0 (0%) |
| Alvarez-Falconi P, et al. [39] | 2010 | 38 (90.5%) | 38 (90.5%) | 31 (73.8%) | 26 (61.9%) | 19 (45.2%) | 11 (26.2%) | 13 (31%) | 15 (35.7%) | 4 (9.5%) | 27 (64.3%) | 23 (54.8%) | 1 (2.4%) | 1 (2.4%) | 19 (45.2%) | 1 (2.4%) | 16 (38.1%) |

NR: Not Reported.

**Table 3. Pooled prevalence of clinical characteristics of Peruvian patients with Oropouche.**

| | Studies | Cases | Sample size | I² (%) | p-value | Prevalence % (95% CI) | Supplementary material |
|---|---|---|---|---|---|---|---|
| Fever | 6 | 390 | 396 | 64% | p = 0.02 | 99 (97 - 100) | S1 Fig |
| Headache. | 6 | 338 | 396 | 48% | p = 0.09 | 86 (80 - 91) | S2 Fig |
| Myalgia. | 5 | 253 | 328 | 27% | p = 0.24 | 77 (71 - 83) | S3 Fig |
| Arthralgia. | 6 | 277 | 396 | 48% | p = 0.09 | 70 (63 - 76) | S4 Fig |
| Anorexia/ Hyporexia. | 5 | 197 | 328 | 69% | p = 0.01 | 56 (45 - 67) | S5 Fig |
| Retroocular pain. | 6 | 211 | 396 | 77% | p < 0.01 | 51 (40 - 62) | S6 Fig |
| Abdominal pain. | 4 | 21 | 231 | 87% | p < 0.01 | 8 (00 - 24) | S7 Fig |
| Nausea/ Vomiting. | 5 | 137 | 299 | 16% | p = 0.32 | 45 (39 - 52) | S8 Fig |
| Diarrhea. | 3 | 23 | 122 | 64% | p = 0.06 | 15 (05 - 30) | S9 Fig |
| Chills. | 4 | 91 | 253 | 99% | p < 0.01 | 40 (00 - 96) | S10 Fig |
| Lumbar pain. | 3 | 105 | 219 | 53% | p = 0.12 | 47 (37 - 58) | S11 Fig |
| Odynophagia. | 5 | 108 | 328 | 92% | p < 0.01 | 21 (07 - 40) | S12 Fig |
| Cutaneous rash. | 5 | 50 | 265 | 83% | p < 0.01 | 14 (04 - 26) | S13 Fig |
| Conjunctival injection. | 3 | 20 | 185 | 96% | p < 0.01 | 09 (00 - 50) | S14 Fig |
| Petechiae. | 4 | 3 | 316 | 0% | p = 0.75 | 01 (00 - 02) | S15 Fig |
| Cough. | 4 | 32 | 253 | 96% | p < 0.01 | 10 (00 - 38) | S16 Fig |

Confidence interval (CI).

closely resemble the findings of a broader global meta-analysis on the clinical presentation of OROV, which reported the following pooled prevalence of symptoms: fever (100%), headache (95%), myalgia (72%), arthralgia (58%), chills (26%), dizziness (69%), weakness/fatigue (45%), and neck/back pain (48%) [4].

Another meta-analysis that included 28 studies and assessed 4,196 patients reported similar results. The most common clinical manifestations of OROV were fever (97%), headache (86.5%), myalgia (72.3%), general malaise or fatigue (56.4%), arthralgia (50.3%), chills (49.6%), loss of appetite (44.3%), eye pain (43.2%), and back pain (31.7%) [42]. A systematic review compared the symptoms of Oropouche with those of dengue and other arboviral diseases, finding no differences in the frequency of fever or headaches between patients with Oropouche and those with dengue. However, odynophagia and abdominal pain were more common in Oropouche patients than in dengue patients (OR of 3.20 and 2.50, respectively). On the other hand, myalgia and arthralgia were less frequent in Oropouche patients compared to those with dengue [10].

Peru is an endemic country for arboviral diseases, such as dengue, Zika, chikungunya, and Oropouche, which have a significant impact on public health [11,32]. In this context, public health faces significant challenges. The geographical expansion of OROV in Peru, which covers regions such as Piura, Loreto, Huánuco, Madre de Dios, and San Martín, suggests a spread of the virus beyond its traditional endemic areas, likely influenced by factors such as ecosystem disturbances, population movements, and changes in land use [43–45]. A shift in vector distribution patterns, particularly with *Culicoides* species being more widely spread due to changing climate conditions, has significantly contributed to this geographical expansion, complicating efforts to control the virus [25,46].

A systematic review suggests a potential shift in the epidemiological status of the disease, raising questions about which vectors are driving transmission. In this context, two mosquito species, *Culicoides paraensis* and *C. sonorensis*, consistently demonstrate high competence in transmitting OROV, with transmission rates around 30%. In contrast, other mosquito species, including *Aedes* and *Culex spp.*, consistently showed infection rates below 20%, indicating limited OROV transmission [47].

Recent studies have reported an increase in OROV cases in the Brazilian Amazon region, attributed to environmental factors and potential genetic changes in the virus [24]. This recent surge is linked to a new reassortant strain of OROV,

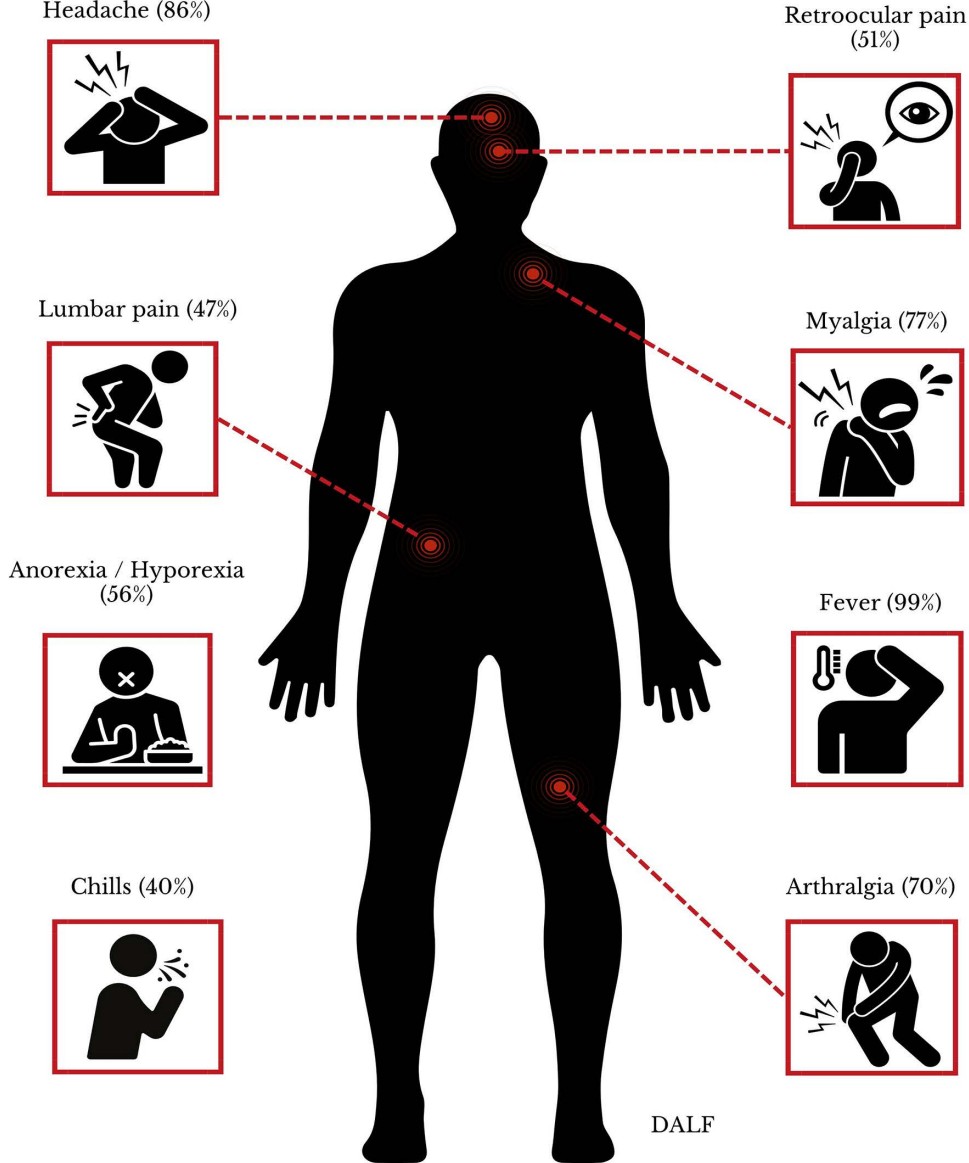

**Fig 3. Most prevalent clinical manifestations in Peruvian patients with Oropouche.**

which combines genetic segments from strains in the eastern Amazon and neighboring countries such as Peru, Colombia, and Ecuador. This new strain exhibits higher replication and virulence, reducing the effectiveness of pre-existing immunity in previously infected individuals. Furthermore, the virus's spread is not only due to the movement of vectors but also to human migration, facilitating its expansion in the region [24,48,49].

On the other hand, the fact that most cases are concentrated in adults between the ages of 20 and 39 aligns with previous research, which may be explained by greater exposure to vectors in occupational or recreational settings within forested areas [50]. Regarding the sex distribution, although more cases are reported in men (44.9%) compared to women (32.8%), this difference should be analyzed with caution, considering the lack of complete information in some studies and potential biases in data collection.

In all the studies analyzed, patients affected by the OROV showed a positive clinical evolution, with no reported fatalities, which aligns with the existing literature that considers this infection to be mild and self-limiting. However, it is important to note that, although rare, neurological complications such as meningitis and meningoencephalitis have been reported in studies conducted outside of Peru [51]. Additionally, some recent studies have highlighted the possibility of symptom recurrence in a significant percentage of cases, reaching up to 60%. This underscores the importance of conducting more prolonged clinical follow-up in future research [52].

The notable variation in the prevalence of certain symptoms (I² > 75%) may be attributed to several factors, including the diversity of geographic areas involved and discrepancies in diagnostic methods used, such as RT-PCR versus ELISA [29]. This highlights the urgent need to strengthen and ensure proper training for healthcare personnel in vulnerable areas, where the simultaneous circulation of various arboviruses is common. Moreover, the overlap in symptoms between diseases such as dengue and Oropouche further complicates accurate diagnosis, increasing the risk of misdiagnosis and subdiagnosis. Given the increasing spread of Oropouche alongside other arboviruses, it is imperative to implement more refined diagnostic protocols and improve surveillance systems to effectively differentiate between these diseases and ensure timely, accurate medical responses [53,54].

On the other hand, an assessment of the quality of the included studies revealed that all were of moderate quality, with the studies being observational in nature (retrospective or cross-sectional). This finding is consistent with the results of the global meta-analysis on Oropouche fever, which also noted that the majority of studies were retrospective and lacked high-quality data [4].

This study presents several strengths. First, it is the first meta-analysis focused exclusively on the clinical manifestations of OROV in the Peruvian population, providing valuable information on this virus in Latin America, an area with limited knowledge. Second, it offers robust evidence to differentiate OROV clinical manifestations from other recently emerging diseases in Peru, such as dengue and mpox [32,33,55]. Furthermore, the study rigorously follows PRISMA guidelines, ensuring the reproducibility of results, and uses multiple international and regional databases to ensure a thorough and comprehensive search.

Finally, this study opens the door for the generation of new knowledge through future research. These could include comparisons between the clinical manifestations of OROV and other arboviruses, such as dengue, Zika, and chikungunya, as well as the evaluation of coinfections with these diseases [10,56,57]. Given the prevalence of OROV in Peru, studies could also be conducted to assess the population's knowledge and attitude towards this disease [58]. Additionally, the study could explore the public's willingness to receive a vaccine against Oropouche if available and establish prevention measures [59]. It would also be valuable to develop information dissemination strategies regarding OROV through reliable media channels, following the model used in other international outbreaks [60].

The study has several key limitations. First, the small number of included studies (n = 6) limits the generalizability of the findings and may affect the precision of the estimates. Second, the inability to assess publication bias due to the limited number of studies could introduce undetected bias. Additionally, most of the studies are retrospective, which may lead to biases such as missing data or unmeasured confounding variables. The heterogeneity among the studies in terms of design and patient characteristics may have also influenced the findings, making it difficult to identify consistent patterns. Lastly, the wide confidence intervals in some estimates indicate high uncertainty, requiring caution in interpretation and highlighting the need for more studies with larger sample sizes and more accurate measurements.

## 5. Conclusions

This study sheds light on the clinical manifestations of OROV, which share significant symptoms with other arboviruses, complicating diagnosis. This overlap highlights the critical need for precise diagnosis, as clinical confusion often leads to underestimating the disease. Differentiating between OROV and similar febrile diseases, such as dengue, remains a challenge. The growing number of cases and geographic spread emphasizes the urgency of public health responses.

To enhance control and response, strengthening epidemiological surveillance is vital, along with the development of more sensitive diagnostic tools. Public health interventions should target endemic regions to mitigate the spread of OROV. Additionally, further research into OROV's virology, epidemiology, and immune response is essential. Efforts should also focus on developing vaccines and antiviral therapies to combat the emerging infection.

Given the reemergence of OROV and increased cases, its inclusion in clinical algorithms for managing acute febrile syndrome in endemic areas is recommended. Future research should address subclinical presentations and the economic/social impacts. A multidisciplinary approach is necessary to contain the virus and mitigate its effects on vulnerable communities. Educational campaigns on prevention and management should be prioritized to reduce incidence and improve public health.

## Supporting information

**S1 Table. PRISMA Checklist (PRISMA 2020 Main Checklist and PRIMSA Abstract Checklist).**
(DOCX)

**S2 Table. The adjusted search terms as per searched electronic databases or search tools.**
(DOCX)

**S3 Table. Quality of the studies included in the systematic review and meta-analysis.**
(DOCX)

**S4 Table. Database.**
(XLSX)

**S5 Table. Table of excluded studies.**
(DOCX)

**S6 Table. Meta-analysis database.**
(DOCX)

**S7 Table. R version 4.2.3. script.**
(DOCX)

**S1 Fig. Pooled prevalence of fever in Peruvian patients with Oropouche.**
(TIF)

**S2 Fig. Pooled prevalence of headache in Peruvian patients with Oropouche.**
(TIF)

**S3 Fig. Pooled prevalence of myalgia in Peruvian patients with Oropouche.**
(TIF)

**S4 Fig. Pooled prevalence of arthralgia in Peruvian patients with Oropouche.**
(TIF)

**S5 Fig. Pooled prevalence of anorexia/ hyporexia in Peruvian patients with Oropouche.**
(TIF)

**S6 Fig. Pooled prevalence of retroocular pain in Peruvian patients with Oropouche.**
(TIF)

**S7 Fig. Pooled prevalence of abdominal pain in Peruvian patients with Oropouche.**
(TIF)

**S8 Fig. Pooled prevalence of nausea/vomiting in Peruvian patients with Oropouche.**
(TIF)

**S9 Fig. Pooled prevalence of diarrhea in Peruvian patients with Oropouche.**
(TIF)

**S10 Fig. Pooled prevalence of chills in Peruvian patients with Oropouche.**
(TIF)

**S11 Fig. Pooled prevalence of lumbar pain in Peruvian patients with Oropouche.**
(TIF)

**S12 Fig. Pooled prevalence of odynophagia in Peruvian patients with Oropouche.**
(TIF)

**S13 Fig. Pooled prevalence of cutaneous rash in Peruvian patients with Oropouche.**
(TIF)

**S14 Fig. Pooled prevalence of conjunctival injection in Peruvian patients with Oropouche.**
(TIF)

**S15 Fig. Pooled prevalence of petechiae in Peruvian patients with Oropouche.**
(TIF)

**S16 Fig. Pooled prevalence of cough in Peruvian patients with Oropouche.**
(TIF)

## Author contributions

**Conceptualization:** Darwin A. León-Figueroa, Edwin Aguirre-Milachay, Milagros Diaz-Torres, Jean Pierre Villanueva-De La Cruz, Mario J. Valladares-Garrido.

**Data curation:** Darwin A. León-Figueroa, Edwin Aguirre-Milachay, Milagros Diaz-Torres, Jean Pierre Villanueva-De La Cruz, Edwin A. Garcia-Vasquez, Virgilio E. Failoc-Rojas, Mario J. Valladares-Garrido.

**Formal analysis:** Darwin A. León-Figueroa, Edwin Aguirre-Milachay, Edwin A. Garcia-Vasquez, Virgilio E. Failoc-Rojas, Mario J. Valladares-Garrido.

**Investigation:** Darwin A. León-Figueroa, Edwin Aguirre-Milachay, Milagros Diaz-Torres, Jean Pierre Villanueva-De La Cruz, Edwin A. Garcia-Vasquez, Virgilio E. Failoc-Rojas, Mario J. Valladares-Garrido.

**Methodology:** Darwin A. León-Figueroa, Edwin Aguirre-Milachay, Milagros Diaz-Torres, Jean Pierre Villanueva-De La Cruz, Edwin A. Garcia-Vasquez, Virgilio E. Failoc-Rojas, Mario J. Valladares-Garrido.

**Project administration:** Darwin A. León-Figueroa, Jean Pierre Villanueva-De La Cruz, Edwin A. Garcia-Vasquez, Virgilio E. Failoc-Rojas.

**Resources:** Darwin A. León-Figueroa, Edwin Aguirre-Milachay, Milagros Diaz-Torres, Jean Pierre Villanueva-De La Cruz, Edwin A. Garcia-Vasquez, Virgilio E. Failoc-Rojas.

**Software:** Darwin A. León-Figueroa, Milagros Diaz-Torres, Virgilio E. Failoc-Rojas, Mario J. Valladares-Garrido.

**Supervision:** Darwin A. León-Figueroa, Edwin Aguirre-Milachay, Milagros Diaz-Torres, Edwin A. Garcia-Vasquez, Virgilio E. Failoc-Rojas, Mario J. Valladares-Garrido.

**Validation:** Darwin A. León-Figueroa, Edwin Aguirre-Milachay, Milagros Diaz-Torres, Mario J. Valladares-Garrido.

**Visualization:** Darwin A. León-Figueroa, Edwin Aguirre-Milachay, Jean Pierre Villanueva-De La Cruz, Edwin A. Garcia-Vasquez, Virgilio E. Failoc-Rojas, Mario J. Valladares-Garrido.

**Writing – original draft:** Darwin A. León-Figueroa, Edwin Aguirre-Milachay, Milagros Diaz-Torres, Jean Pierre Villanueva-De La Cruz, Edwin A. Garcia-Vasquez, Virgilio E. Failoc-Rojas, Mario J. Valladares-Garrido.

**Writing – review & editing:** Darwin A. León-Figueroa, Milagros Diaz-Torres, Jean Pierre Villanueva-De La Cruz, Edwin A. Garcia-Vasquez, Virgilio E. Failoc-Rojas, Mario J. Valladares-Garrido.

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
