## [Decision Letter · Decision Letter 0]

24 Sep 2025

Dear Dr. Valladares-Garrido,

The authors need to answer the comments of the reviewers and clarify the role of the vector in Oropouche virus transmission and the possibility of confusion between infection of DENV-3 and OROV symptoms.

We look forward to receiving your revised manuscript.

Kind regards,

Victoria Pando-Robles, Ph.D.

Academic Editor

PLOS ONE

Journal Requirements:

2. Please remove all personal information, ensure that the data shared are in accordance with participant consent, and re-upload a fully anonymized data set.

Additional guidance on preparing raw data for publication can be found in our Data Policy (https://journals.plos.org/plosone/s/data-availability#loc-human-research-participant-data-and-other-sensitive-data) and in the following article: http://www.bmj.com/content/340/bmj.c181.long .

Reviewers' comments:

Reviewer's Responses to Questions

**Comments to the Author**

1. Is the manuscript technically sound, and do the data support the conclusions?

Reviewer #1: Yes

Reviewer #2: Partly

2. Has the statistical analysis been performed appropriately and rigorously?

Reviewer #1: Yes

Reviewer #2: N/A

3. Have the authors made all data underlying the findings in their manuscript fully available?

Reviewer #1: Yes

Reviewer #2: Yes

4. Is the manuscript presented in an intelligible fashion and written in standard English?

Reviewer #1: Yes

Reviewer #2: Yes

Reviewer #1: The manuscript by León-Figueroa and colleagues addresses an increasingly worrisome pathogen from South American. The manuscript is exceptionally well written and uses standard techniques for reviewing the literature, assessing quality of relevant reports and performing statistical evaluations. I view this manuscript as a valuable addition to the literature on OROV and do not have substantive critisms. I do have two rather trivial editorial suggestions:

Line 43: Oropouche virus (OROV) -> OROV (you already defined OROV on line 33)

Lines 56-60: I would suggest you delete all the %, CI and I^2 values from the abstract since they are presented in Table 2, and simply state “The most common symptoms were fever, headache, …” to enhance readability of the abstract.

Reviewer #2: A more detailed description of the overlap between vectors and cases should be provided. It is also necessary to address the changes in symptoms observed when DENV-3 was introduced into the region. The data are valuable, but they need to be presented in a more accessible and reader-friendly manner.

**Do you want your identity to be public for this peer review?** For information about this choice, including consent withdrawal, please see our Privacy Policy

Reviewer #1: No

Reviewer #2: **Yes: ** Ma Isabel Salazar

---

## [Author Response · Author response to Decision Letter 1]

3 Oct 2025

Reviewer #1:

1. Reviewer says: The manuscript by León-Figueroa and colleagues addresses an increasingly worrisome pathogen from South American. The manuscript is exceptionally well written and uses standard techniques for reviewing the literature, assessing quality of relevant reports and performing statistical evaluations. I view this manuscript as a valuable addition to the literature on OROV and do not have substantive critisms. I do have two rather trivial editorial suggestions:

Our response: We deeply appreciate your valuable comments and observations on our manuscript. Your recommendations have not only been essential in improving the quality of the information presented but have also strengthened our research, providing clarity and precision to the points discussed. We are pleased to know that the manuscript has been well received, and we will continue to work diligently to ensure that it meets the highest standards of quality.

2. Reviewer says: Line 43: Oropouche virus (OROV) -> OROV (you already defined OROV on line 33)

Our response: The term OROV was corrected and reviewed throughout the article, ensuring consistency.

3. Reviewer says: Lines 56-60: I would suggest you delete all the %, CI and I^2 values from the abstract since they are presented in Table 2, and simply state “The most common symptoms were fever, headache, …” to enhance readability of the abstract.

Our response: The suggested values were removed from the abstract, which has significantly improved its readability.

Reviewer #2:

Thank you for your valuable feedback and for providing constructive comments that have significantly improved the quality and clarity of our manuscript. Below is a detailed explanation of how we have addressed the observations provided:

1. Reviewer says: A more detailed description of the overlap between vectors and cases should be provided.

Our response: We have improved the introduction and discussion sections to provide a more comprehensive understanding of the vectors involved in Oropouche virus (OROV) transmission. Specifically, we have expanded on the diversity of vectors transmitting OROV, including not only Culicoides paraensis but also Aedes aegypti, Aedes albopictus, and Culex quinquefasciatus, which are becoming increasingly relevant in urban settings with high population density.

In response to the reviewer's comments regarding the overlap between vectors, we have incorporated recent studies that highlight the co-circulation of OROV with other arboviruses such as dengue. These studies suggest that the presence of multiple vectors, such as Aedes and Culicoides species, increases the complexity of transmission dynamics, potentially leading to higher morbidity and complications in diagnosis.

Additionally, we have clarified the impact of environmental factors on vector populations, noting that rising temperatures and changing precipitation patterns have created more favorable breeding conditions for Culicoides mosquitoes. Recent studies have shown how climate change can affect the geographical spread of these vectors, further increasing the risk of OROV outbreaks.

2. Reviewer says: It is also necessary to address the changes in symptoms observed when DENV-3 was introduced into the region.

Our response: We agree with the reviewer’s suggestion to address the potential confusion between the symptoms of Oropouche virus (OROV) infections and dengue virus, particularly in regions where both diseases are endemic. In the manuscript, we have added a section that discusses the overlapping clinical manifestations of OROV and dengue, such as fever, headache, and myalgia. These similarities make differential diagnosis challenging, especially in areas with high co-circulation of multiple mosquito-borne viruses.

Additionally, we have incorporated recent findings that show how the introduction of DENV-3 in areas affected by OROV has influenced the clinical presentation of Oropouche fever. However, the discussion of dengue and other arboviruses, such as Zika and Chikungunya, has been addressed in a general way. In Peru, DENV-2 is more common, and when it occurs as a secondary infection, it tends to be more lethal with a worse prognosis. On the other hand, DENV-3, in studies where it occurs as a primary infection, is associated with a higher risk of complications.

3. Reviewer says: The data are valuable, but they need to be presented in a more accessible and reader-friendly manner.

Our response: To enhance the manuscript and address the reviewer’s request for a more accessible and reader-friendly presentation, we have added new references from recent studies that explore the interaction between OROV and other arboviruses, the role of climate change in vector distribution, and the diagnostic challenges posed by co-infections. These references not only support our discussion but also provide additional context for understanding the evolving epidemiology of OROV.

If you have any comments or recommendations, we are ready to respond.

Sincerely,

Mario J. Valladares-Garrido

Escuela de Medicina Humana, Universidad Señor de Sipán, Chiclayo, Peru; vgarri-do@uss.edu.pe

---

## [Decision Letter · Decision Letter 1]

27 Oct 2025

Dear Dr. Valladares-Garrido,

Thank you for submitting your manuscript to PLOS ONE. After careful consideration, we feel that it has merit but does not fully meet PLOS ONE’s publication criteria as it currently stands. Therefore, we invite you to submit a revised version of the manuscript that addresses the points raised during the review process.

To proceed with publication, I would appreciate it if you would take into account the reviewers' recent comments.

We look forward to receiving your revised manuscript.

Kind regards,

Victoria Pando-Robles, Ph.D.

Academic Editor

PLOS ONE

Journal Requirements:

Reviewers' comments:

Reviewer's Responses to Questions

**Comments to the Author**

Reviewer #1: All comments have been addressed

Reviewer #3: (No Response)

2. Is the manuscript technically sound, and do the data support the conclusions?

Reviewer #1: Yes

Reviewer #3: Yes

3. Has the statistical analysis been performed appropriately and rigorously?

Reviewer #1: Yes

Reviewer #3: I Don't Know

4. Have the authors made all data underlying the findings in their manuscript fully available?

Reviewer #1: Yes

Reviewer #3: Yes

5. Is the manuscript presented in an intelligible fashion and written in standard English?

Reviewer #1: Yes

Reviewer #3: Yes

Reviewer #1: THe changes you made certainly enhance the clarity of several aspects of the manuscript. Very nice job.

Reviewer #3: The manuscript “Oropouche infection in Peruvian patients: a systematic review and meta-analysis.” presents an interesting analysis of the epidemiological profile of OROV in Peru. I have some comments and suggestions before making a final decision.

-> Introduction

- Highlight deaths from OROV in Brazil. This finding is unprecedented;

- Regarding the differential diagnosis between dengue and Oropouche, this is a very important issue and some studies have been dedicated to this topic. I suggest reading and mentioning some, such as doi:10.1016/j.lana.2024.100718. e doi: 10.1186/s12985-025-02945-x;

- There is a multifactorial discussion about the increase in OROV cases/circulation. In addition to environmental issues, some authors have discussed important changes in the viral genome. I suggest reading and citing some studies, such as doi: 10.1038/s41591-024-03300-3, doi: 10.1101/2024.07.27.24310296 e doi: 10.1002/jmv.70112;

Figure 1: Write the viral taxa in italics; dengue and chikungunya do not start with a capital letter; clarify what the numbers after the virus names mean (confirmed cases?); to what time interval do the cases correspond?

I don't understand why the manuscript's strengths and limitations are referenced and why they are the same references -> This study presents several strengths [10,32,33]; The study has several key limitations [10,32,33].

**Do you want your identity to be public for this peer review?** For information about this choice, including consent withdrawal, please see our Privacy Policy

Reviewer #1: No

Reviewer #3: No

---

## [Author Response · Author response to Decision Letter 2]

2 Nov 2025

Reviewer #1:

1. Reviewer says: The changes you made certainly enhance the clarity of several aspects of the manuscript. Very nice job.

Our response: Thank you very much for your review. Your comments and suggestions have been crucial in improving the quality of our article.

Reviewer #3:

1. The manuscript “Oropouche infection in Peruvian patients: a systematic review and meta-analysis.” presents an interesting analysis of the epidemiological profile of OROV in Peru. I have some comments and suggestions before making a final decision.

Our response: Thank you for your thoughtful review and for recognizing the value of our analysis. We appreciate your comments and suggestions, and we will carefully address them to further improve the manuscript. We look forward to making the necessary revisions and submitting an updated version for your final consideration.

-> Introduction

2. Reviewer says: Highlight deaths from OROV in Brazil. This finding is unprecedented.

Our response: Thank you very much for your valuable comments and suggestions. I have made the necessary revisions to highlight the fatal cases of Oropouche virus (OROV) in Brazil, as requested. In the revised version, I have emphasized the unprecedented finding of these deaths, marking a significant increase in the severity of the virus's impact in non-endemic regions such as Brazil. This observation has been included to underline the urgent need for attention due to the potential evolution of more virulent strains.

Thank you again for your insightful feedback, which has been very helpful in improving the clarity and precision of the manuscript.

3. Reviewer says: Regarding the differential diagnosis between dengue and Oropouche, this is a very important issue and some studies have been dedicated to this topic. I suggest reading and mentioning some, such as doi:10.1016/j.lana.2024.100718. e doi: 10.1186/s12985-025-02945-x.

Our response: Thank you for the valuable recommendation. I have incorporated the suggested information, highlighting the urgent need to strengthen healthcare personnel training in vulnerable areas where the simultaneous circulation of various arboviruses is common. Additionally, I have emphasized that the overlap in symptoms between diseases such as dengue and Oropouche further complicates accurate diagnosis, increasing the risk of misdiagnosis and underdiagnosis. In this context, given the increasing spread of Oropouche alongside other arboviruses, I believe it is essential to implement more refined diagnostic protocols and improve surveillance systems to effectively differentiate these diseases and ensure timely and accurate medical responses.

4. Reviewer says: There is a multifactorial discussion about the increase in OROV cases/circulation. In addition to environmental issues, some authors have discussed important changes in the viral genome. I suggest reading and citing some studies, such as doi: 10.1038/s41591-024-03300-3, doi: 10.1101/2024.07.27.24310296 e doi: 10.1002/jmv.70112.

Our response: Thank you for your observation. I have reviewed your comment, and I agree that the discussion regarding the increase in OROV cases should address both environmental factors and potential genetic changes in the virus. In my text, I included information about a new reassortant OROV strain that has emerged in the Brazilian Amazon region, which combines genetic segments from strains in the eastern Amazon and neighboring countries such as Peru, Colombia, and Ecuador. This strain has shown higher replication and virulence, contributing to a reduction in the effectiveness of pre-existing immunity in previously infected individuals. Additionally, I highlighted the role of human migration and the movement of vectors as facilitating factors for the virus's spread in the region, which is also considered in recent literature.

5. Reviewer says: Figure 1: Write the viral taxa in italics; dengue and chikungunya do not start with a capital letter; clarify what the numbers after the virus names mean (confirmed cases?); to what time interval do the cases correspond?.

Our response: Thank you very much for your valuable suggestions. The corrections have been made to Figure 1, including the italicization of the viral taxa names, the correction of the disease names with a lowercase initial letter, and the clarification that the numbers represent confirmed cases, with the corresponding time interval specified (Epidemiological Week 38 of 2025).

Once again, thank you for your feedback, and I remain available for any further comments.

6. Reviewer says: I don't understand why the manuscript's strengths and limitations are referenced and why they are the same references -> This study presents several strengths [10,32,33]; The study has several key limitations [10,32,33].

Our response: Thank you for your observation. We have removed the references from the indicated sections. The context arose because these references pertained to other meta-analyses that shared common strengths and limitations. However, we have made the correction as per your suggestion

If you have any comments or recommendations, we are ready to respond.

Sincerely,

Mario J. Valladares-Garrido

Escuela de Medicina Humana, Universidad Señor de Sipán, Chiclayo, Peru; vgarri-do@uss.edu.pe

---

## [Decision Letter · Decision Letter 2]

11 Nov 2025

Oropouche infection in Peruvian patients: a systematic review and meta-analysis.

PONE-D-25-43147R2

Dear Dr. Valladares-Garrido,

We’re pleased to inform you that your manuscript has been judged scientifically suitable for publication and will be formally accepted for publication once it meets all outstanding technical requirements.

Kind regards,

Victoria Pando-Robles, Ph.D.

Academic Editor

PLOS ONE

Additional Editor Comments (optional):

Reviewers' comments:

Reviewer's Responses to Questions

**Comments to the Author**

Reviewer #1: All comments have been addressed

Reviewer #3: All comments have been addressed

2. Is the manuscript technically sound, and do the data support the conclusions?

Reviewer #1: Yes

Reviewer #3: Yes

3. Has the statistical analysis been performed appropriately and rigorously?

Reviewer #1: Yes

Reviewer #3: I Don't Know

4. Have the authors made all data underlying the findings in their manuscript fully available?

Reviewer #1: Yes

Reviewer #3: Yes

5. Is the manuscript presented in an intelligible fashion and written in standard English?

Reviewer #1: Yes

Reviewer #3: Yes

Reviewer #1: The authors have done a very credible job in addressing reviewer comments and I continue to judge this as a valuable contribution to the field.

Reviewer #3: The authors addressed my main concerns, suggestions, and comments. The manuscript is suitable for publication.

**Do you want your identity to be public for this peer review?** For information about this choice, including consent withdrawal, please see our Privacy Policy

Reviewer #1: No

Reviewer #3: No

---

## [Editor Report · Acceptance letter]

PONE-D-25-43147R2

PLOS ONE

Dear Dr. Valladares-Garrido,

I'm pleased to inform you that your manuscript has been deemed suitable for publication in PLOS ONE. Congratulations! Your manuscript is now being handed over to our production team.

Kind regards,

on behalf of

Victoria Pando-Robles

Academic Editor

PLOS ONE